# How the Potassium Channel Response of T Lymphocytes to the Tumor Microenvironment Shapes Antitumor Immunity

**DOI:** 10.3390/cancers14153564

**Published:** 2022-07-22

**Authors:** Martina Chirra, Hannah S. Newton, Vaibhavkumar S. Gawali, Trisha M. Wise-Draper, Ameet A. Chimote, Laura Conforti

**Affiliations:** 1Division of Nephrology, Department of Internal Medicine, University of Cincinnati, Cincinnati, OH 45267, USA; chirrama@ucmail.uc.edu (M.C.); hannah.newton@nih.gov (H.S.N.); vaibhavkumar.gawali@crl.com (V.S.G.); ameet.chimote@uc.edu (A.A.C.); 2Division of Hematology-Oncology, Department of Internal Medicine, University of Cincinnati, Cincinnati, OH 45267, USA; wiseth@ucmail.uc.edu

**Keywords:** ion channels, potassium channels, antitumor immunity, tumor microenvironment

## Abstract

**Simple Summary:**

Ion channels are proteins that control the movement of ions across the membranes of cells, thus regulating their physiological functions. In T lymphocytes, ion channels control the Ca^2+^ influx that, in turn, promotes proliferation and effector functions. In the context of cancer, T cells have the role of fighting tumor cells. However, the tumor microenvironment negatively regulates T cell antitumor capabilities. Therefore, it is of utmost importance to understand the relationship between the tumor microenvironment and the ion channel apparatus of T cells to overcome the tumors’ immunosuppressive capabilities.

**Abstract:**

Competent antitumor immune cells are fundamental for tumor surveillance and combating active cancers. Once established, tumors generate a tumor microenvironment (TME) consisting of complex cellular and metabolic elements that serve to suppress the function of antitumor immune cells. T lymphocytes are key cellular elements of the TME. In this review, we explore the role of ion channels, particularly K^+^ channels, in mediating the suppressive effects of the TME on T cells. First, we will review the complex network of ion channels that mediate Ca^2+^ influx and control effector functions in T cells. Then, we will discuss how multiple features of the TME influence the antitumor capabilities of T cells via ion channels. We will focus on hypoxia, adenosine, and ionic imbalances in the TME, as well as overexpression of programmed cell death ligand 1 by cancer cells that either suppress K^+^ channels in T cells and/or benefit from regulating these channels’ activity, ultimately shaping the immune response. Finally, we will review some of the cancer treatment implications related to ion channels. A better understanding of the effects of the TME on ion channels in T lymphocytes could promote the development of more effective immunotherapies, especially for resistant solid malignancies.

## 1. Introduction

The immune system is essential for cancer surveillance. However, it is recognized that the unique milieu associated with tumors (known as the tumor microenvironment—TME) is responsible for the functional abnormalities of the immune system that impair cancer elimination [1,2]. The cellular components of the TME (which include fibroblasts, vascular, and immune cells) together with extracellular, soluble, and metabolic features inhibit the functions of a complex network of antitumor immune cells that would otherwise prevent tumor growth and metastasis [3,4]. Effector T cells are key players in the fight against cancer, and the presence of cytotoxic CD8^+^ and Th1 CD4^+^ T cells in the tumor is associated with a favorable prognosis and response to immunotherapies. However, the aforementioned alterations in the TME negatively affect the T cell capacity to infiltrate the tumors and kill cancer cells, which, to this day, remain the rate-limiting steps towards an effective antitumor response [5]. In order for T cells to infiltrate the tumor and exert their antitumor functions, it is imperative that the appropriate movements of ions across plasma and intracellular membranes occur, as T cells are critically dependent on ion channel regulated-Ca^2+^ signaling for their function [6].

This review focuses on the role that ion channels, particularly of the potassium (K^+^) class, play in mediating the immunosuppressive effects of the TME on T cells. Following a brief overview of the current understanding of the role of ion channels in normal T cell function, we will discuss the negative impact of the metabolic (hypoxia, adenosine, and electrolyte imbalance) and cellular (programmed cell death ligand 1, PD-L1) components of the TME on T cell immune surveillance and their dependence on K^+^ channels. Furthermore, we will highlight the important role that K^+^ channels play in response to immune checkpoint inhibitors and the potential benefits of developing new single agent or combination cancer immunotherapies targeting K^+^ channels.

## 2. Ion Channel Network in T lymphocytes

Ion channels and transporters form an intricate network responsible for intracellular ion homeostasis and Ca^2+^ levels necessary for effector functions in T lymphocytes (Figure 1) [7,8]. Intracellular Ca^2+^, in fact, regulates the activation of Ca^2+^-dependent transcription factors, the expression of most activation genes in T cells, and mechanical functions such as exocytosis and motility [9,10,11]. The expression profile and role of different ion channels in T cell function have been described in detail by others [7,8,12]. Briefly, there are five major ion channels in human T cells, with each playing a unique and key role in ion homeostasis: Ca^2+^ release-activated Ca^2+^ (CRAC) channels; two K^+^ channels, the voltage-gated K^+^ channel Kv1.3 and the Ca^2+^ dependent K^+^ channel KCa3.1; and transient receptor potential (TRP) channels (e.g., TRPM4 and TRPM7).

The first of these ion channels that play a direct role in controlling intracellular Ca^2+^ levels are the CRAC channels. CRAC channels allow the influx of Ca^2+^ after T cell receptor (TCR) stimulation [8] (Figure 1). These channels are formed by two different subunits, ORAI1 (ORAI calcium release-activated calcium modulator 1)—and/or its homologs ORAI2 and ORAI3—and STIM1/STIM2 (stromal interaction molecule 1/2) accessory proteins [6]. ORAI subunits represent the pore-forming subunits of the CRAC channel and are localized in the plasma membrane. Conversely, STIM proteins are on the endoplasmic reticulum (ER) membrane and play a key role as Ca^2+^-sensors. The contribution of these channels to the T cell response to antigen presentation is the following: antigen presentation and recognition at the TCR/CD3 complex activates phospholipase Cγ (PLCγ), which cleaves phosphatidylinositol 4,5-bisphosphate (PIP_2_) into inositol (1,4,5)—trisphosphate (IP_3_) and diacylglycerol (DAG) (Figure 1). IP_3_ then binds to its receptor located on the ER, resulting in the release of Ca^2+^ from the ER store. The STIM subunits detect the Ca^2+^ depletion from the ER, oligomerize and migrate next to the plasma membrane, where they interact with ORAI, leading to its opening and, consequently, Ca^2+^ influx into the T cell (store-operated calcium entry—SOCE) [7,8,13,14]. Early studies in human and murine T cells revealed that the production of cytokines, such as interleukin-2 (IL-2), interleukin-4 (IL-4), and interferon-γ (IFN-γ), and granule exocytosis required SOCE through STIM1 and ORAI1 [11,15,16]. Furthermore, Weidinger and colleagues (2013) showed that the genetic deletion of STIM1 and STIM2 in cytotoxic T cells in mice reduced cancer cell killing [17]. An increase in intracellular Ca^2+^ regulates important transcription factors such as nuclear factor of activated T cells (NF-AT) and nuclear factor kappa B (NF-κB), which control the transcription of multiple genes. It has been estimated that more than 80% of T cell activation genes are regulated by Ca^2+^ [9]. The specificity of the transcription factor activated by Ca^2+^ influx depends on the shape and amplitude of the Ca^2+^ response [18].

Kv1.3 and KCa3.1, the principal K^+^ channels in human T cells, play a significant role in Ca^2+^ influx by maintaining the electrochemical driving force necessary for Ca^2+^ entry into the cell. The depolarization of the cell following CRAC-mediated Ca^2+^ influx activates the voltage-dependent Kv1.3, while the increase in intracellular Ca^2+^ levels activates KCa3.1, a Ca^2+^-activated voltage-independent K^+^ channel. The subsequent increase in K^+^ efflux through these two K^+^ channels hyperpolarizes the plasma membrane and sustains the influx of the positively charged Ca^2+^ ions through the CRAC channels [7,8,19]. Conversely, activation of Na^+^ influx and Cl^−^ efflux, and the inhibition of K^+^ efflux, all of which lead to membrane depolarization, negatively regulate CRAC-mediated Ca^2+^ fluxes [7].

Lastly, TRP channels represent non-selective cation channels and are categorized into six subfamilies (TRPA, TRPC, TRPV, TRPM, TRPP, TRPML) [20]. In T cells, TRPM7 channels allow Ca^2+^ and Mg^2+^ influx through the plasma membrane and are involved in cell development and motility. In contrast, TRPM4 allows Na^+^ influx, thus contributing to the depolarization of the membrane potential and reducing the driving force for Ca^2+^ entry [20,21].

Overall, each ion channel plays a significant role in a complex network that ultimately shapes the effector functions of T cells.

## 3. Potassium Channels and T Cell Functions

Ion channels mediate T cell differentiation and T cell functions such as proliferation, motility, cytokine release and cytotoxicity, and are implicated in multiple pathological conditions including infections [22,23], autoimmunity [24,25,26,27,28,29,30,31], and cancer [32,33].

Specific to K^+^ channels, their quantitative expression, and, consequently, their functional role, varies depending on the T cell activation status (resting versus activated) and subset (naïve—T_naïve_, central memory—T_CM_, and effector memory—T_EM_ T cells) [7,8,19,31] In general, resting human T cells predominantly express Kv1.3 compared to KCa3.1 channels. Upon activation, the type of K^+^ channel that is preferentially upregulated depends on the T cell subset. Activated T_EM_ cells (CD45RO^+^CCR7^−^) express high levels of Kv1.3 channels, while KCa3.1 channels are more abundant in activated naïve (CD45RO^−^CCR7^+^) and T_CM_ cells (CD45RO^+^CCR7^+^) [7,8,19,31]. Functionally, a seminal work by Wulff et al. in 2003 showed that autoantigen-specific T_EM_ cells overexpressed Kv1.3, and their proliferative capacity was selectively suppressed by Kv1.3 blockers [34]. A potential therapeutic role for Kv1.3 channel blockers has been reported for many diseases where autoreactive T_EM_ cells contribute to disease pathology including rheumatoid arthritis, type-1 diabetes mellitus and systemic lupus erythematosus (SLE) [27,35]. The functional capabilities of T_naïve_ and T_CM_ are instead suppressed by KCa3.1 blockers which have been shown to be effective in animal models of inflammatory bowel disease and asthma [36]. Interestingly, while repeatedly stimulated antigen-specific T cells that acquire a T_EM_ phenotype progressively overexpress Kv1.3 and rely on it, and not KCa3.1, for their effector functions, in case of loss of Kv1.3 function, KCa3.1 has been shown to compensate, maintaining the proliferation and the effector responses of the T cells, though not to the full extent [37].

Overall, both K^+^ channels are implicated in proliferation and cytokine production. Kv1.3 and KCa3.1 blockade suppress the proliferation of human T cells [7,38]. Conversely, highly proliferative tumor-infiltrating lymphocytes (TILs), characterized by the high expression of the proliferation marker Ki67, express high levels of Kv1.3 [32]. Blockade of Kv1.3 and KCa3.1 channels also suppresses the production of cytokines, including those with antitumor functions such as IFN-γ, tumor necrosis factor- α (TNF-α) and IL-2, in different subsets of effector T cells [39,40]. Kv1.3 channels are also implicated in T cell differentiation and cytotoxicity. Kv1.3 channels regulate the differentiation of CD8^+^ T cells, as evidenced by the fact that the expression of a dominant-negative Kv1.3 mutant decreased the differentiation of activated CD8^+^ T cells into T_EM_ cells and led to the reversion of T_EM_ cells into the T_CM_ phenotype [29]. This is true also for CD4^+^ T cells, in which Kv1.3 loss-of-function mutations have shown to (a) negatively impact CD4^+^ T_CM_ conversion into T_EM_ cells, (b) lead to reversion of CD4^+^ T_EM_ into T_CM_ cells, and c) slow down CD4^+^ T_EM_ cell proliferation and differentiation by blocking the G2/M phase of the cell cycle [41]. Knockdown of Kv1.3 in T cells has also been reported to shift the T cell population from a memory to a naïve phenotype [39]. Furthermore, Kv1.3 channels regulate T cell cytotoxicity. Hu and colleagues (2013) showed that Kv1.3 channels regulate granzyme B (GrB) release early upon activation [29]. However, at later time points, KCa3.1 blockade also suppressed GrB levels [29]. Studies of Kv1.3 channels in CD8^+^ TILs of head and neck cancer patients showed that Kv1.3 expression correlated with GrB levels and marked functionally competent T cells [32]. The inhibitory effect of K^+^ channel blockade on cytokine production/release and cytotoxicity is a consequence of their regulatory function on SOCE.

Ion channels also play a major role in the homing and migration of effector T cells. Effective leukocyte migration is fundamental to combatting infections, as shown in primary immunodeficiency caused by defects in leukocyte motility and trafficking, and cancer [42]. In cancer, the close proximity between effector cytotoxic CD8^+^ T cells and cancer cells allows physical contact between the two cells and facilitates the direct killing of tumor cells. K^+^ and TRP channels have emerged as crucial components as they are involved in the control of cell motility regulators such as membrane potential and cellular volume [21,43].

In migrating human T cells, KCa3.1, together with TRPM7 channels, compartmentalize at the uropod, and their interplay controls the forward motion of these cells, possibly through the concerted regulation of membrane potential, Ca^2+^ influx, and actomyosin contractility (Figure 2) [21]. Oscillations in intracellular Ca^2+^ levels were reported to occur exclusively at the uropod, revealing an intracellular Ca^2+^ gradient that may be necessary to support opposing cellular events occurring at the two poles of a migrating cell. In contrast to KCa3.1 and TRPM7, Kv1.3 and ORAI1 are confined to the leading-edge, and their blockade does not influence T cell migration [21]. However, the position of Kv1.3 and CRAC channels at the leading-edge may facilitate the interaction with the antigen-presenting cells and the activation process itself [21]. Others have reported in an in vivo mouse model that CRAC channels oversee activated human T lymphocytes homing to lymph nodes [44]. The abolishment of CRAC activity suppressed CCL12—a chemokine responsible for T cell homing—activation of integrins required for trans-endothelial migration and lymph node entry of T lymphocytes [44]. However, in naïve CD4^+^ T cells, genetic inactivation of STIM1 (or STIM1/STIM2) did not impair homing of T cells to secondary lymphoid organs in an in vivo model [45]. CRAC-mediated Ca^2+^ influxes also controlled the timing of the “stop and go” behavior of T cells during the search for the cognate antigen, and the decrease in velocity upon cognate antigen stimulation, an advantageous feature for effective contact with the antigen presenting cell (APC) [45,46]. Indeed, a dominant-negative ORAI1 mutant and CRAC blockade caused an increase in cell velocity, an inefficient strategy during the search for the cognate antigen that could affect antigen recognition at the tumor site [21,46].

While KCa3.1 channels play an important role in integrin- and chemokine-driven migration, they do not seem to regulate trans-endothelial migration [21,33,40,47]. Indeed, Sim and colleagues (2017) showed that a KCa3.1 inhibitor reduced the migration of human CD8^+^ T_EM_ cells that express high levels of interleukin (IL)-7 receptor alpha (IL-7Rα) chain on intercellular adhesion molecule 1 (ICAM1) surfaces, but not trans-endothelial migration [40]. There is also evidence that Kv1.3 plays a role in the motility of rat T cells [48]. Using a rat model of skin delayed-type hypersensitivity, the authors showed that, while the Kv1.3 blockade did not prevent the accumulation of activated antigen-specific CD4^+^ T_EM_ cells at the site of inflammation, once in the inflamed area it caused the T_EM_ cells to have unproductive contact with antigen-bearing APCs, insufficient re-activation, and inefficient Ca^2+^-dependent activation of ß1 integrin, thus decreasing motility and velocity [48].

Overall, K^+^ channels are regulators of key T cell functions that are necessary for an effective antitumor response. These include the ability of the cells to chemotax and infiltrate the tumor, make contact with cancer cells, as well as deliver the “lethal hit” and produce antitumor cytokines. However, T cells fail to perform these functions in cancer as they encounter a hostile TME.

## 4. Tumor Microenvironment and Ion Channels

Multiple elements of the TME affect T cell antitumor capabilities through alterations of K^+^ homeostasis (Figure 3).

### 4.1. Hypoxia

Proliferating tumor cells require and consume more oxygen (O_2_) than that provided by pre-existing blood vessels, decreasing the O_2_ availability for other cell types. While the tumor may stimulate increased vascularization, this adaptation is typically dysfunctional to sustain adequate O_2_ concentrations with newly formed blood vessels that are leaky, with blind ends and temporary occlusions [49]. The resulting hypoxia affects both cancer and immune cell function. Levels of O_2_ tension as low as ≤2.5 mmHg have been measured in tumors [50]. The effects of hypoxia occur both transcriptionally (through the stabilization of the well-described transcription factor HIF1α, hypoxia-inducible factor 1α) and post-transcriptionally. The induction of HIF1α causes cancer cells to undergo metabolic reprogramming, which leads to an increased glucose uptake with the production of lactate, even in the presence of O_2_ and functional mitochondria (termed the “Warburg effect”) [51,52]. These metabolic changes support cancer cell growth in hypoxic regions and drive tumor advantage for migration, invasion, metastases, and resistance to treatments [49,53]. Hypoxic effects have also been shown to occur through alterations in the function of voltage-dependent K^+^ channels in cancer cells such as PC12 pheochromocytoma cells, and other cells of tissues that respond quickly to change in O_2_ availability such as carotid body cells and pulmonary artery smooth muscle cells [54].

Likewise, hypoxia alters the immune system. Hypoxia decreases T cell activation, cytotoxicity, and cytokine release [55]. Hypoxia delays cytotoxic T cell development in vitro and decreases T cell activation in vivo [55,56]. In CD4^+^ T cells, hypoxia induces the differentiation of CD4^+^ T cells into regulatory T (T_reg_) or T helper 17 cells [52]. While hypoxia has these effects on immune cells, it additionally alters the TME by promoting the up-regulation of immune checkpoint inhibitors, such as T-lymphocyte–associated antigen 4 (CTLA-4) and PD-L1 [52] and inducing acidosis and the production of adenosine [4].

Overall, the presence of hypoxia in tumors has been associated with poor prognosis and response to therapies. The overall detrimental effects of hypoxia in cancer are exemplified by the results of a study on the effects of hyperoxia in mice [57]. In this study, the authors showed that a reduction in intratumoral hypoxia via the delivery of O_2_ to tumor-bearing mice converted an immunosuppressive TME into an immunopermissive one [57]. Increasing O_2_ availability in the tumor reduced extracellular adenosine, increased T cell infiltration and release of immune stimulating cytokines and chemokines, and decreased tumor growth factor β (TGFβ). These effects were entirely dependent on T and NK cells. It also reduced T_reg_ accumulation in the tumor and CTLA-4 expression in these cells. Ultimately, it improved the regression of tumors, prevented spontaneous metastasis, and enhanced long-term survival. Furthermore, it improved the efficacy of adoptively transferred tumor-reactive CD8^+^ T cells [57].

Kv1.3 channels play an important role as transducers of O_2_ deprivation in T cells. Indeed, hypoxia inhibits Kv1.3 channels in T cells in both the acute (minutes) and chronic (days) setting [58,59]. These inhibitions occurred post-transcriptionally, with no changes in Kv1.3 mRNA. Acute hypoxia inhibited Kv1.3 currents in human T cells and Jurkat T cells [58] caused membrane depolarization and, consequently, the suppression of TCR-mediated Ca^2+^ influx (Figure 3) [59]. These effects of hypoxia on Ca^2+^ fluxes were mainly attributable to Kv1.3, because there was no change in CRAC and Ca^2+^-activated K^+^ channel function [59]. The src protein tyrosine kinase p56Lck was required for Kv1.3 response to hypoxia [60]. Supporting this conclusion, p56Lck-deficient cell lines and human T lymphocytes pre-treated with a src protein tyrosine kinase inhibitor lost Kv1.3-mediated response to hypoxia [60]. Chronic hypoxia instead induced a decrease in the pore-forming Kv1.3α subunit protein expression in Jurkat T cells and, consequently, a decrease in functional Kv1.3 channels [58]. Notably, in sustained O_2_ depletion, Kv1.3 current inhibition was also associated with reduced T cell proliferation induced by TCR cross-linking agents [58]. Experiments focused on the underlying mechanisms that led to this decrease in Kv1.3 channel expression during chronic exposure to hypoxia in T cells [61] and revealed that chronic hypoxia decreased Kv1.3 surface expression by inhibiting the forward trafficking of Kv1.3 from the trans-Golgi to the plasma membrane, which physiologically involved clathrin-coated vesicle formation initiated via the AP1 adaptor protein (Figure 3). Under hypoxic conditions, gene and protein expression of the γ subunit of AP1 is decreased with the consequent inhibition of clathrin-coated vesicle formation and impaired forward trafficking of newly synthesized Kv1.3 to the plasma membrane [61]. Together, these results provide evidence of a role of Kv1.3 channels in the T cell response to changes in O_2_ tension, thus explaining the linkage between hypoxia, T cell immunosuppression, and the ion channel apparatus.

### 4.2. Necrosis and Ionic Imbalance

Despite hypoxic conditions and the inadequate provision of nutrients, cancer cells continue to rapidly divide, thus creating conditions for irregular areas of necrosis within the tumor mass. Eil and colleagues (2016) demonstrated that these regions of tumor necrosis lead to extracellular ionic imbalance as the dying cells release their intracellular contents and, thus, high levels of K^+^ ions [62]. The intracellular [K^+^] is 154 mM, much higher than the extracellular [K^+^] (5.4 mM). This increase in extracellular [K^+^] in the necrotic tumor areas ultimately leads to increased intracellular [K^+^] in the TILs through pump or leak channels, due to the attenuation of K^+^ chemical gradient (Figure 3) [62]. No alteration in membrane potential was reported. However, this ionic imbalance inhibited the serine/threonine kinases Akt and mTOR (mammalian target of rapamycin) in the TCR-signaling pathway, ultimately suppressing T cell transcriptional programs required for cytokine production (Figure 3) [62]. This finding is in line with the fact that necrotic areas represent a marker of poor prognosis in a variety of solid malignant tumors [63,64]. Interestingly, Kv1.3 overexpression corrected the immunosuppressive K^+^ accumulation in tumor necrotic areas, increased IL-2 and IFN-γ production, reduced tumor burden and prolonged survival of melanoma bearing mice [62]. KCa3.1 activation via specific pharmacological activators also rescued T cell function in high extracellular [K^+^] in vitro [62]. The effect of KCa3.1 activation in vivo has yet to be determined. Recently, Ong and colleagues (2019) showed that human peripheral blood T cells were negatively influenced by high extracellular [K^+^] [65]. Indeed, high extracellular [K^+^] and the consequent high intracellular [K^+^] led to impaired T cell proliferation in T_CM_ and T_EM_, but not in T memory stem (T_SCM_) cells. Moreover, high extracellular [K^+^] suppressed T cell cytokine production and tumor cell killing ability and enhanced programmed cell death protein-1 (PD-1) expression [65]. This suppression of T cell function in high extracellular [K^+^] was accelerated by the blockade of KCa3.1 and Kv1.3, while the activation of KCa3.1 restored T cell function [65].

Therefore, it may be a fair assumption to predict that the immunosuppressive effects of K^+^ accumulation in necrotic areas could be exacerbated by the concomitant presence of hypoxia, adenosine and PD-L1 (see below) in the TME that suppress K^+^ channels.

Overall, these data show the importance of K^+^ channels in correcting the immunosuppressive effects induced by necrosis. However, we should also take into account that in a later study, Vodnala and colleagues (2019) showed that the increase in extracellular [K^+^] within the tumor has an additional impact on T cell stemness [66]. They demonstrated that extracellular [K^+^] leads to T cell starvation and metabolic and epigenetic reprogramming that, while limiting T cell effector functions, promote stem cell-like programs [66].

### 4.3. Adenosine

Adenosine, a catabolic product of adenosine triphosphate (ATP), is an anti-inflammatory nucleoside that represents one of the predominant negative immune regulators of the TME. Tumors accumulate extracellular ATP through passive loss from dead cells and inflammatory- or hypoxia-induced release from stressed cells [67,68]. This accumulation of ATP leads to increased levels of adenosine as the ecto-nucleotidases CD39 and CD73 metabolize ATP to adenosine monophosphate (AMP) and AMP to adenosine, respectively (Figure 3) [67,68]. These enzymes are present on the surface of adenosine-producing cells such as stromal cells, tumor cells, or T_reg_ that, in addition, can also directly secrete adenosine through bidirectional nucleoside transporters [68]. Extracellular levels of adenosine in solid tumors can be as high as 10 μM; >300 times higher than those found in healthy tissues (30 nM) [69,70]. Adenosine can then bind to its receptors on the surface of T cells and exert its anti-inflammatory functions, which include the inhibition of effector cells and the promotion of suppressive T_reg_ [68,71]. Extracellular adenosine is decreased through degradation via adenosine deaminase or transport into cells by the cell membrane nucleoside transporters. However, in the TME, salvage pathways can be inhibited, thereby adding to the extracellular adenosine concentration [68,69]. Hypoxia facilitates the accumulation of adenosine in the TME by increasing the expression of adenosine-producing enzymes and reducing the expression of adenosine-degrading enzymes via HIF1α [4,69].

There is strong evidence that the immunosuppressive effects of adenosine on effector T cells are partly mediated through the inhibition of KCa3.1 channels. In activated human T cells, adenosine binds to A_2A_ receptor (A_2A_R), increases cAMP production, and, consequently, induces protein kinase A1 (PKAI) activation. These changes serve to inhibit KCa3.1 and cause decreased T cell integrin-dependent motility, chemotaxis, and cytokine release (Figure 3) [33,47]. Additionally, KCa3.1 channels mediate the sensitivity to adenosine of circulating T cells from head and neck squamous cell carcinoma (HNSCC) patients [33]. It is well established that the immunosuppressive effects of adenosine are more pronounced in T cells from cancer patients than healthy donors [33,71]. More specifically and relevant to KCa3.1, chemotaxis was significantly more inhibited by adenosine in HNSCC CD8^+^ T cells than in CD8^+^ T cells from healthy controls [33]. This effect of adenosine in HNSCC CD8^+^ T cells was neither mediated by an increase in A_2A_R expression nor proximal cAMP-PKAI signaling. Rather, a reduction in KCa3.1 functionality (and not expression) in HNSCC CD8^+^ T cells as compared to healthy CD8^+^ T cells accounted for the increased sensitivity to adenosine and translated into reduced infiltration into the adenosine-rich HNSCC tumors [33]. Indeed, patients whose circulating CD8^+^ chemotaxis was most severely inhibited by adenosine were also those who had the lowest infiltration of CD8^+^ T cells into the tumor [33]. Important from a therapeutic perspective, positive modulators of KCa3.1 restored the chemotaxis of HNSCC CD8^+^ T cells in the presence of adenosine [33]. More recently, follow-up studies showed that the reduction in KCa3.1 functionality in HNSCC CD8^+^ T cells was secondary to a defect in plasma-membrane localized calmodulin, a signaling molecule tethered to the channels’ C-terminal domain and responsible for its Ca^2+^-sensitivity [72]. Binding of Ca^2+^ to calmodulin translates into a conformational change in KCa3.1 and its opening [73]. Knockdown of calmodulin in healthy CD8^+^ T cells reproduced HNSCC T cell dysfunction, namely reducing KCa3.1 activity and chemotactic capabilities in the presence of adenosine [33,72]. In contrast, pre-incubation of calmodulin knocked-down healthy CD8^+^ T cells with a specific KCa3.1 activator (1-EBIO) rescued the cells’ capability to chemotax in the presence of adenosine [72]. Therefore, KCa3.1 activators could be used to overcome the immunosuppressive action of adenosine and improve the antitumor functions of T cells.

The blockade of the adenosine pathway and the subsequent increase in antitumor immune functions may be also achieved by targeting different points along the adenosine pathway [68]. These include inhibiting adenosine synthesis [74,75], increasing adenosine degradation through increased adenosine deaminase activity [71], or preventing the interaction with specific receptors on the surface of target cells [76]. We have shown that targeted silencing of A_2A_R in memory CD45RO^+^CD8^+^ T cell of HNSCC patients via lipid nanoparticles abrogated the inhibitory effect of adenosine on the chemotaxis of these cells [77]. In vivo studies provide conclusive evidence of the importance of adenosine in cancer [76,78]. A phase I clinical trial of treatment-refractory renal cell cancer patients demonstrated antitumor activity of an A_2A_R antagonist alone and in combination with checkpoint inhibitors and established the safety and feasibility of targeting this pathway [76]. However, the advantage of KCa3.1 activators over conventional adenosine receptor blockers is that this treatment approach may overcome the actions of prostaglandin E2 (PGE2), another immunosuppressive molecule that accumulates in the TME and inhibits KCa3.1. It has been reported that PGE2 inhibits KCa3.1 in mast cells [79].

Overall, the development of immunotherapies targeting KCa3.1 on the adenosine pathway may serve as promising therapeutics [76,80].

### 4.4. Programmed Death Ligand 1

Overexpression of PD-L1 is utilized by cancer cells to suppress the antitumor immune response. The effect of PD-L1 on T lymphocytes is mediated by engagement with its cognate receptor PD-1 and, consequently, inhibition of TCR-dependent effector functions, such as Ca^2+^ fluxing, secretion of cytokines and cytotoxicity [81]. KCa3.1 channels provide a link between PD-1 stimulation by PD-L1 and reduced Ca^2+^-related functions [82]. PD-L1 reduces KCa3.1 activity and Ca^2+^ fluxes in CD8^+^ T cells; this effect is rescued by treatment with anti-PD-1 blocking antibodies. KCa3.1 function is controlled by multiple mechanisms, including the regulation of the channel Ca^2+^ sensitivity by calmodulin (described before) and histidine phosphorylation [83,84]. Histidine phosphorylation of KCa3.1 has been shown to be triggered by the phosphoinositide 3-kinase (PI3K): the phosphatidylinositol-3 phosphatase (PI3P) signaling pathway [84,85]. In T cells, ligation of PD-1 by PD-L1 results in a decrease in TCR proximal signaling through inhibition of PI3K activity [86]. A decrease in PI3K suppresses the production of PI3P from phosphatidylinositol (PI). PI3P is known to activate the nucleoside diphosphate kinase B (NDPK-B), which, ultimately, increases KCa3.1 activity via histidine phosphorylation [84,85]. Thus, the reduction in PI3K triggered by PD-1 stimulation suppresses KCa3.1 (Figure 3). Indeed, we showed that the response of KCa3.1 channels to PD-1 ligation is mediated by PI3K-PI3P signaling after short exposure to PD-L1. Longer exposures to PD-L1 show a role for calmodulin. Five-day exposure to PD-L1 reduced calmodulin expression in T cells by 40% while the contribution of PI3K was reduced compared to early time points [82]. The involvement of KCa3.1 in mediating the effect of the PD-L1/PD-1 response is further supported by in vitro studies showing that PD-1 and PD-L1 blocking antibodies increased KCa3.1 activity [82]. An increase in Kv1.3 activity was also reported. However, there may be a difference in sensitivity to PD-L1 between these two channels; a higher concentration of anti-PD-L1 blocking antibodies was necessary to unleash Kv1.3 activity in T cells from cancer patients compared to KCa3.1 [82]. Further studies are necessary to dissect the mechanisms mediating the effect of PD-L1 on Kv1.3 channels. Ex vivo studies of K^+^ channel activity in CD8^+^ T lymphocytes of HNSCC patients treated with anti-PD-1 blocking antibodies also showed an increase in KCa3.1 and Kv1.3 activities [87]. These studies position K^+^ channels downstream to immune checkpoint inhibitors.

Immune checkpoint inhibitors have arisen as new effective treatments in solid malignancies as they have shown remarkable improvements in treatment outcomes including long-lasting remissions after discontinuation of therapy [88,89]. However, the majority of patients do not respond or eventually relapse [90]. Activation of K^+^ channels may be critical to expanding the efficacy of these revolutionary treatments.

Kv1.3 channels have also been recently implicated as biomarkers of response to immunotherapy. Goggi and colleagues (2022) developed a new Kv1.3 targeting radiopharmaceutical ([18F]AlF-NOTA-KCNA3P) and showed that it could be used to differentiate tumors responding to immune checkpoint inhibitors (anti-PD-1 antibodies alone or in combination with anti-CTLA-4 antibodies) in a syngeneic colon cancer model [91]. This radiolabeled ligand of Kv1.3 allowed the quantification of T_EM_ cells infiltrating the tumor, which were increased in responding tumors, and raised the possibility that a similar strategy could be used to develop new biomarkers of response to immune checkpoint inhibitors.

## 5. K^+^ Channels as a Target for Cancer Therapy

We and others have examined the effects of the TME on the ion channel apparatus of antitumor immune cells (Table 1) and have explored the potential of restoring impaired channel function for cancer treatment. As multiple elements of the TME affect K^+^ channels in T cells as described above, the anticipated repercussion of this would be reduced K^+^ channel activity and Ca^2+^ fluxes in TILs. Indeed, the Ca^2+^ response of TILs is severely dampened compared to circulating T cells [32,92]. In HNSCC patients, TILs exhibit suppressed Kv1.3 and KCa3.1 function compared to circulating T cells. Additionally, we showed that the number of Kv1.3 channels per membrane unit surface (described as Kv1.3 current density) was lower in HNSCC TILs compared to HNSCC PBT. Hence, TILs had a significantly lower number of channels. [32,87]. Moreover, TILs were not only “functionally” but also “spatially” impaired. The expression of Kv1.3 was shown to be higher in stromal than epithelial CD8^+^ TILs, and highly proliferating-cytotoxic TILs were physically excluded from the tumor and trapped in the stroma [32]. In this scenario, ion channels represent a fascinating target to counteract the immunosuppressive effects of the TME on immune cells and to restore antitumor immunity. Proof of the benefits of targeting Kv1.3 in cancer in vivo have already been provided, while the benefits of KCa3.1 are still indirect, as they are supported only by in vitro evidence [33,62]. On the other side, ion channels are currently investigated as therapeutic strategies in autoimmune diseases, the opposite situation compared to cancer, whereas the immune system is hyperactive rather than suppressed. As a consequence, the immune system’s hyperactivity in autoimmune disorders may be prevented by the inhibition of ion fluxes, which are responsible for the functionality of disease-provoking immune cells [93,94]. In proof of this, our group has showed that silencing RNA against Kv1.3 in memory T cells improved survival in a mouse model of SLE [35]. Thus, upregulating K^+^ channel function in T cells, and particularly Kv1.3 in the context of cancer, while very promising, has to take into consideration the potential development of immune-related adverse events that are already reported for the immune checkpoint inhibitors currently in use [95].

Recently, we have investigated the effects of the immune checkpoint inhibitor pembrolizumab (a monoclonal antibody against PD-1) on the ion channel functionality in T cells of HNSCC patients [87]. These studies revealed that patients who responded to neoadjuvant pembrolizumab (where the response was defined according to the % of viable tumor at resection, namely less than 80%) had a specific ion channel signature in circulating T cells. In more detail, while in TILs pembrolizumab induced an increase in Kv1.3 activity and Ca^2+^ fluxes regardless of the response, a characteristic response was identified in circulating T cells [87]. Interestingly, all patients (responders and non-responders) experienced an immediate increase in KCa3.1 activity in circulating CD8^+^ T cells after pembrolizumab administration. However, circulating CD8^+^ T cells of patients who responded to pembrolizumab also had a marked increase in Kv1.3 activity compared with non-responders. This was accompanied by an increase in Ca^2+^ fluxes and chemotactic ability in the presence of the immunosuppressive adenosine [87]. Hence, K^+^ channels are significantly implicated in the response to pembrolizumab in HNSCC patients, where responders are characterized by a typical ion channel signature (Kv1.3^high^ KCa3.1^high^ Ca^2+high^) in circulating cytotoxic T cells that may also elucidate strategies to overcome resistance.

However, the network of ion channels that contribute to Ca^2+^ fluxes in T cells is complex. As discussed above, ion channels are differentially expressed in resting and activated T cells, and their expression depends on the T cell subset. Moreover, many different settings (e.g., tumors, autoimmune and infectious diseases) and models (e.g., rat, mouse, and human) were used to study ion channel properties and, therefore, the translatability between fields may not be completely compatible and, thus, may add to the heterogeneity of results. This is true particularly for studies on Kv1.3 channels that utilize mouse T cells, as the expression and functional role of Kv1.3 channels is different in mice compared to human T cells [96]. Additionally, the ion channel network is a very sensitive and highly regulated system. For example, T cells operate efficiently towards cancer elimination in a defined and relatively small range of extracellular [Ca^2+^]: between 23 and 625 µM, which allows the release of an elevated number of cytotoxic granules per cell [97].

Nevertheless, K^+^ channels contribute to the reduction in the antitumor function of T cells and targeting these channels has shown promising therapeutic value. As K^+^ channels are downstream to multiple immunosuppressive elements of the TME, targeting these channels would ultimately allow to simultaneously counteract the multidirectional attacks employed by the tumor to suppress antitumor immunity.

## 6. Conclusions

Ion channels play an integral role in Ca^2+^ signaling and downstream T cell effector functions, and, therefore, are, when fully functional, critically important for the immune system role in detecting and eliminating cancer cells. Although this ion channel network is complex, it is clear that many factors within the TME converge to impair ion channel function (Figure 3). Further understanding the effect of the TME on ion channels could help to develop effective treatments, especially for patients resistant to conventional immunotherapies.

## Figures and Tables

**Figure 1 cancers-14-03564-f001:**
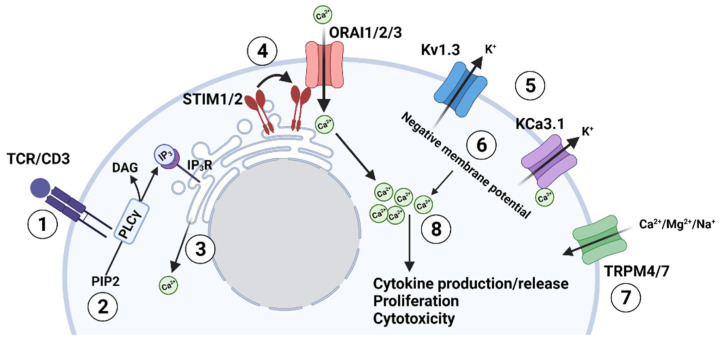
TCR-mediated regulation of the ion channel network in T lymphocytes. (1) Antigen stimulates the T cell receptor (TCR); (2) TCR stimulation activates phospholipase C (PLCγ), which cleaves phosphatidylinositol 4,5-bisphosphate (PIP_2_) to inositol (1,4,5)-trisphosphate (IP_3_) and diacylglycerol (DAG); (3) IP_3_ binds to its cognate receptor (IP_3_R) on the endoplasmic reticulum (ER) and depletes the ER Ca^2+^ store; (4) The depletion of Ca^2+^ in the ER causes the STIM protein (located on the ER membrane) to oligomerize and associate with the ORAI subunits (located on the plasma membrane) forming a functional CRAC channel (Ca^2+^ release-activated Ca^2+^ channel) that allows the influx of Ca^2+^ into the cell; (5) The increase in intracellular Ca^2+^ depolarizes the cell thus activating the voltage-gated K^+^ channel Kv1.3, which allows the efflux of K^+^ ions. While at the same time, the increase in intracellular Ca^2+^ activates the Ca^2+^ dependent K^+^ channel KCa3.1, which also allows the efflux of potassium ions; (6) The efflux of K^+^ ions hyperpolarizes the membrane; (7) TRP channels can also play a role in Ca^2+^ influx directly (like TRPM7) or by regulating the membrane potential (like TRPM4); (8) The negative membrane potential ultimately supports the electrochemical gradient necessary for the sustained influx of Ca^2+^ through the CRAC channels and, consequently, the increase intracellular Ca^2+^ levels needed for downstream functions such as cytokine production or release, T cell proliferation and cytotoxicity.

**Figure 2 cancers-14-03564-f002:**
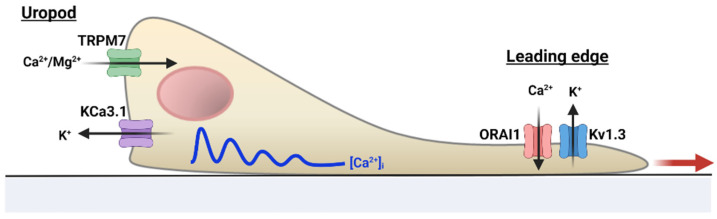
Ion channel regulation of human T cell motility. Ion channels acquire a distinct polarization in migrating human T cells: KCa3.1 and TRPM7 localize at the uropod, while Kv1.3 and CRAC (ORAI1) are at the leading-edge. Intracellular Ca^2+^ levels ([Ca^2+^]_i_) are higher at the uropod than other cell compartments, which may serve to support the Ca^2+^ requirements for cell motility. The blue line indicates intracellular Ca^2+^ oscillations; the red arrow indicates directionality of movement.

**Figure 3 cancers-14-03564-f003:**
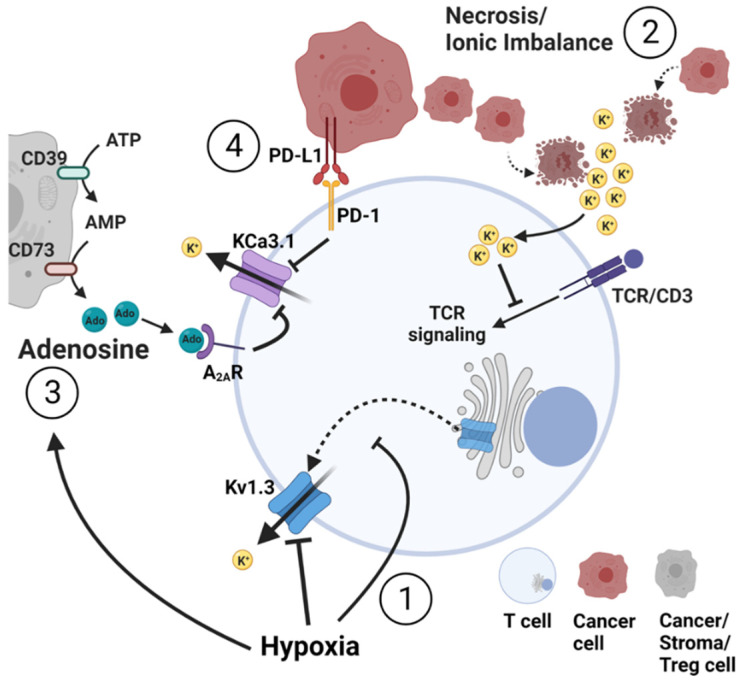
Effect of the tumor microenvironment (TME) on K^+^ homeostasis in T lymphocytes. (1) Hypoxia inhibits Kv1.3 channels via a dual mechanism. Short-term exposure to hypoxia directly inhibits Kv1.3 activity by changing the channel’s phosphorylation state. Prolonged exposure to low O_2_ tension reduces the forward trafficking of newly synthesized channel’s pore-forming subunits from the trans-Golgi to the plasma membrane. (2) Necrosis or cell death leads to ionic imbalances and increased K^+^ ions in the extracellular space. The consequent increase in intracellular K^+^ inhibits TCR activation via blockade of TCR signaling. (3) Adenosine (Ado) accumulates in the extracellular space via consecutive degradation of ATP and AMP by the nucleotidase CD39 and CD73 present on the membrane of cancer, stroma and T_reg_ cells. Adenosine binds to its receptor on T cells and triggers KCa3.1 inhibition via cAMP and PKA1 (4) PD-L1 presented by cancer cells binds to the PD-1 receptor on the T cells and triggers KCa3.1 inhibition through PI3K signaling and calmodulin. Inhibition of Kv1.3 and KCa3.1 channels by hypoxia, adenosine and PD-L1 (1,3,4) decreases overall T cell effector functions. Accumulation of K^+^ in the cells and its deleterious effects on T cell function can be exacerbated by the concomitant presence of hypoxia, adenosine and PD-L1 in the TME. Conversely, overexpression and/or activation of Kv1.3 and KCa3.1 channels can simultaneously counteract all these immunosuppressive elements of the TME.

**Table 1 cancers-14-03564-t001:** Effects of the tumor microenvironment (TME) characteristics on K^+^ channel network of T cells. Abbreviations: TILs, tumor-infiltrating lymphocytes; PD-L1, programmed cell death ligand 1; PD-1, programmed cell death protein-1.

TME Feature	Effect on Ion Channels	References
Hypoxia		
Acute	Inhibition of Kv1.3 currents	[58,59]
Chronic	Decrease in Kv1.3 expression	[58,61]
Tumor necrosis and increase in intra-tumoral K^+^ concentration	Responsible for the increase in intracellular [K^+^] in TILs that leads to inhibition of T cell transcriptional programs required for cytokine production. Kv1.3 overexpression (in vivo) and overexpression/pharmacological activation of KCa3.1 (in vitro) rescue T cell function	[62]
Adenosine	Adenosine binds to A_2A_ receptor on T cells triggering KCa3.1 inhibition	[33,47,72]
PD-L1	PD-L1 binds to PD-1 receptor on T cells triggering KCa3.1 inhibition	[82]

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
