# Peer review of "How the Potassium Channel Response of T Lymphocytes to the Tumor Microenvironment Shapes Antitumor Immunity"

_cancers, 2022, doi:10.3390/cancers14153564_

Round 1

Reviewer 1 Report

Dr. Chirra et al. comprehensively summarized T-cell specific regulatory network for Ca2+  fluxes coupled with K+ ion efflux mediated with Kv1.3 and KCa3.1 channels in circulating T-cells, T-cells in normal tissues and tumor mass. They also paid many attentions to mention effects of hypoxia, necrosis, adenosine, and the PD-1 - PD-L1 axis on the activity and expression levels of Kv1.3 and KCa3.1 channels in tumor tissues and their underlying mechanisms. This manuscript is helpful for readers who are interested in immunotherapy of cancer and other auto-immunologic diseases. Kv1.3 is one of the voltage-gated K+ channel  family members, which is mainly expressed in T-cells. It will be useful if Dr. Chirra et al. could discuss whether circulating T-cells and TILs express the same set of voltage-gated K+ channel  genes. According to their own work, PD-1 antibody administration could increase the activity of Kv1.3 in circulating T-cells in patients with and without therapy response. But they did not mention the alteration status of Kv1.3 activity in TILs, which may provide some clues for understanding immunotherapy responses. In addition, it has been reported that T-cells in tumor mass maintain the T-cell stemness by increased K+ concentration, but could not normally differentiate to functional T-cells (DOI: 10.1126/science.aau0135). I suggest they add some information on this field.

Author Response

Comment 1: Dr. Chirra et al. comprehensively summarized T-cell specific regulatory network for Ca2+ fluxes coupled with K+ ion efflux mediated with Kv1.3 and KCa3.1 channels in circulating T-cells, T-cells in normal tissues and tumor mass. They also paid many attentions to mention effects of hypoxia, necrosis, adenosine, and the PD-1 - PD-L1 axis on the activity and expression levels of Kv1.3 and KCa3.1 channels in tumor tissues and their underlying mechanisms. This manuscript is helpful for readers who are interested in immunotherapy of cancer and other auto-immunologic diseases. Kv1.3 is one of the voltage-gated K+ channel family members, which is mainly expressed in T-cells.

Response 1: We thank the reviewer for this generous comment.

Comment 2: Kv1.3 is one of the voltage-gated K+ channel family members, which is mainly expressed in T-cells. It will be useful if Dr. Chirra et al. could discuss whether circulating T-cells and TILs express the same set of voltage-gated K+ channel genes.

Response 2: We thank the Reviewer for this observation. Unfortunately, we did not conduct any analysis at the gene level regarding Kv1.3 and other voltage-gated K+ channel genes in tumor infiltrating lymphocytes (TILs) versus peripheral blood T cells (PBT). However, in our previous publication (Chimote, A.A.; Hajdu, P.; Sfyris, A.M.; Gleich, B.N.; Wise-Draper, T.; Casper, K.A.; Conforti, L. Kv1. 3 channels mark functionally competent CD8+ tumor-infiltrating lymphocytes in head and neck cancer. Cancer research 2017, 77, 53-612017), we showed that the number of Kv1.3 channels per membrane unit surface (Kv1.3 current density) was lower in head and neck (HNSCC) TILs compared to HNSCC PBT. Hence, TILs had a significantly lower number of channels. We have now added on section 5. K+ channels as a target for cancer therapy the following sentence: “Additionally, we showed that the number of Kv1.3 channels per membrane unit surface (described as Kv1.3 current density) was lower in HNSCC TILs compared to HNSCC PBT. Hence, TILs had a significantly lower number of channels” (lines: 477-480, reference 32).

Comment 3: According to their own work, PD-1 antibody administration could increase the activity of Kv1.3 in circulating T-cells in patients with and without therapy response. But they did not mention the alteration status of Kv1.3 activity in TILs, which may provide some clues for understanding immunotherapy responses.

Response 3: We thank the Reviewer for highlighting this important component. On section 5. K+ channels as a target for cancer therapy the following sentence describes the alteration induced by PD-1 antibody on Kv1.3 in TILs: “In more detail, while in TILs pembrolizumab induced an increase in Kv1.3 activity and Ca2+ fluxes regardless of the response, a characteristic response was identified in circulating T cells (lines: 503-505, reference 87).

Comment 4: In addition, it has been reported that T-cells in tumor mass maintain the T-cell stemness by increased K+ concentration, but could not normally differentiate to functional T-cells (DOI: 10.1126/science.aau0135). I suggest they add some information on this field.

Response 4: We agree with the Reviewer, and we have now added on section 4.2. Necrosis and Ionic Imbalance the following sentence: “However, we should also take into account that in a later study, Vodnala and colleagues (2019) showed that the increase in extracellular [K+] within the tumor has additional impact on T cell stemness. They demonstrated that extracellular [K+] leads to T cell starvation and metabolic and epigenetic reprogramming that, while limiting T cell effector functions, promote stem cell-like programs (lines: 346-350, reference 66).

Reviewer 2 Report

Reviewer comments:

Comments to the Author

The review article by Dr. Chirra et al., focused on the competent anti-tumor immune cells that are fundamental for both tumor surveillance and combating active cancers. This review highlights upon the role of ion channels and how can be targeted for cancer treatment. Article suggest a better reach advancement in deciphering the importance of ion channels in T lymphocytes could promote the development of more effective immunotherapies, especially for resistant solid malignancies.

The review is very interesting in terms of compiling the literature in an organized manner, and article is for the most part well written, the discussion was logical, and the figures and table provided were comprehensive, well validated and presented clearly.

I have the following minor concerns.

             Please add a table on section 4. Tumor Microenvironment and Ion Channels to summarize literature findings.

             Please undergo a thorough check of the manuscript for typographical and grammatical errors.

Author Response

Comment 1: The review article by Dr. Chirra et al., focused on the competent anti-tumor immune cells that are fundamental for both tumor surveillance and combating active cancers. This review highlights upon the role of ion channels and how can be targeted for cancer treatment. Article suggest a better reach advancement in deciphering the importance of ion channels in T lymphocytes could promote the development of more effective immunotherapies, especially for resistant solid malignancies. The review is very interesting in terms of compiling the literature in an organized manner, and article is for the most part well written, the discussion was logical, and the figures and table provided were comprehensive, well validated and presented clearly.

Response 1: Thank you for your enthusiastic comments.

Comment 2: Please add a table on section 4. Tumor Microenvironment and Ion Channels to summarize literature findings.

Response 2: As suggested by the Reviewer, and we have now added a new table (Table 1) which describes the effects of the tumor microenvironment on Kv1.3 and KCa3.1 with the pertinent references.

Comment 3: Please undergo a thorough check of the manuscript for typographical and grammatical errors.

Response 3: We agree with the Reviewer, and we have checked the manuscript for typographical and grammatical errors thanks to the help of two English native-speaker (Dr. Eric Smith, mentioned in the acknowledgments, and Dr. Newton, first co-author). All the changes are in the tracked version mode. The title has now changed from “How the potassium channel response of T lymphocytes to the tumor microenvironment shapes anti-tumor immunity” to “How the potassium channel response of T lymphocytes to the tumor microenvironment shapes antitumor immunity” (word antitumor without the hyphen)